# An Investigation of Real-Time Robotic Polishing Motion Planning Using a Dynamical System

**Xinqing Wang \***[ID]**, Xin Wang, Zhenyu Yang**[ID] **and Yupeng Zou**[ID]

School of Mechanical and Electrical Engineering, China University of Petroleum (East China), Qingdao 266580, China; wx17866613161@163.com (X.W.); z22040040@s.upc.edu.cn (Z.Y.); zouyupeng@upc.edu.cn (Y.Z.)

\* Correspondence: xqwang@upc.edu.cn

**Abstract:** When addressing the technical challenges of achieving precise force tracking during the local polishing process of polishing robots, controlling the contact state between the robot and the workpiece surface is essential. To this end, a contact motion-planning strategy based on dynamic systems is designed to generate trajectory routes during local polishing. The trajectory simulation of the local modulation dynamic system is achieved through the employment of the support vector regression (SVR) algorithm with a Gaussian kernel, facilitating the learning process. The feasibility and stability of planning local paths are validated using the local modulation dynamic system. To maintain a constant contact force between the end-effector polishing robot and the workpiece, an integral adaptive impedance control strategy is utilized, enabling the robot's compliant control. Subsequently, an experimental system for the polishing robot is constructed in order to verify the effectiveness of the motion-planning system. The experimental results demonstrate that the proposed motion-planning approach is applicable in practical polishing processes, ensuring smooth contact and maintaining the desired contact force when scanning nonlinear surfaces, and thus showcasing stability and practicality.

**Keywords:** collaborative robots; polishing; dynamic system; compliance control





## 1. Introduction

With the advancement of modern industry, the shapes of workpieces to be polished, such as automotive parts and hardware, are becoming increasingly unique, and are often produced in small batches [1]. The use of polishing robots to address the demand for small-batch, customized production significantly enhances efficiency and precision. However, as the complexity of workpiece surfaces increases, higher demands are being placed on the polishing quality and applicability of polishing robots [2,3]. Sometimes, it is necessary to polish specific areas of a workpiece, such as automotive parts. The force control and trajectory planning of the polishing robot are the key factors determining the quality of polishing [4]. Thus, there is a need to develop more precise motion control strategies to adjust the robot's interaction forces and motion trajectories.

The most common approach to controlling the interaction forces of polishing robots is based on impedance control strategies [5]. Dai et al. [6] proposed an impedance matching control strategy based on traditional direct force control and impedance control methods, aiming to mitigate the vibration issues of industrial robots caused by large inertia and low stiffness due to impedance control. Although impedance controllers provide compliant behavior at all stages of contact tasks, their force-tracking capability is limited, mainly due to the degree of understanding of the environment. Another method involves the use of a pneumatic constant-force control device at the end effector. Due to its flexibility, simple control, and low cost, this method has been widely applied in robot deburring, sanding, and polishing tasks. However, the nonlinear hysteresis characteristics of the

pneumatic constant-force systems, from the cylinder input to the force output at the device end, significantly affect the system's force control precision [7]. In comparison, admittance control offers advantages in force tracking. Admittance controllers can filter force errors, generating adjustable position signals [8], and control the contact force during polishing by adjusting the coefficients of the admittance controller. Moreover, it has structural advantages, enabling existing position-controlled robots to perform force control without modifying the robot's internal structure [9].

In addition, passive losses generated during the tracking process are a problem for any controller driven by a time-indexed reference trajectory [10,11]. Mirrazavi et al. [12] used a dynamic system approach and time-invariant control to achieve quick responsiveness and the dynamic re-planning of trajectories. Historically, dynamic systems have been used for controlling trajectories in free space, but an increasing amount of research indicates their applicability in controlling contact states with objects [13–15].

This paper proposes a localized-modulation motion-planning strategy based on the original autonomous dynamic-system (DS) control framework for performing contact tasks on complex workpieces. It also introduces an integral adaptive admittance control law to ensure constant contact force during the polishing process. Experiments are designed to validate the feasibility and practicality of the localized-modulation motion-planning strategy.

## 2. Contact Motion-Planning Strategy Based on Dynamic Systems

### 2.1. Locally Modulated Dynamic Systems

Dynamic systems (DS) typically take state variables as inputs and return the rate of change of those variables. If a DS is not an explicit function of time, it is referred to as a primitive autonomous dynamic system:

$$\dot{x} = f(x) \tag{1}$$

where $x \in \mathbb{R}^3$ is defined as the end-effector position in the Cartesian coordinates of the robot. $f(x) : \mathbb{R}^3 \mapsto \mathbb{R}^3$ is a function that maps the current Cartesian position of the robot system to Cartesian velocity. The primitive autonomous dynamic system can be seen as a velocity vector field, which describes the expected behavior at any given position in space.

Locally modulated dynamic systems (LMDS) reshape the primitive autonomous dynamic system model locally, offering the advantage of preserving the stability of the original model. When reshaping the original autonomous dynamic system $f(x)$ via modulation field $M(x)$, the form of the LM dynamic system is as follows:

$$\dot{x} = g(x) = M(x)f(x) \tag{2}$$

where $M(x) \in \mathbb{R}^{3\times3}$ is a continuous matrix-valued modulation function that modulates the primitive autonomous dynamic system $f(x)$, enabling the desired behavior and reshaping the system as a result. When modulated locally with a full rank, the reshaped dynamic system inherits certain properties of the original autonomous dynamic system. The boundedness and stability of LMDS have been extensively demonstrated [16].

Because rotation matrices are always full-rank and act locally, this paper adopts the rotation matrix modulation of the original autonomous dynamic system. To increase flexibility, velocities in the original autonomous dynamic system can be scaled by multiplying the rotation matrix by a scalar. Let $R(x) \in \mathbb{R}^{3\times3}$ represent the state-dependent rotation matrix, and let $\kappa$ denote the velocity scaling factor. When $\kappa = 1$, the original velocity remains unchanged. Then, a modulation function is constructed to locally rotate and accelerate or decelerate the original autonomous dynamic system, as shown below:

$$M(x) = \kappa R(x) \tag{3}$$

The following function is used to control the influence of modulation, determining which regions of the state space are affected by the original autonomous dynamic system.

$$h(\boldsymbol{x}) = 1 - \tanh(v\Gamma(\boldsymbol{x})) \tag{4}$$

where $\Gamma(\boldsymbol{x})$ represents the distance from any position in space to the surface, and $v$ represents the rotation speed coefficient controlling the rate of decay of the rotation matrix.

Let $w(\boldsymbol{x}) = h(\boldsymbol{x})\theta$ denote the state-dependent rotation, which smooths the rotation of the original autonomous dynamic system, rotating it fully by an angle of $\theta$ only at $\Gamma(\boldsymbol{x}) = 0$.

The modulation function is defined as the state-dependent rotation matrix:

$$\boldsymbol{R}(\boldsymbol{x}) = \boldsymbol{I} + \sin(w(\boldsymbol{x}))\boldsymbol{K} + (1 - \cos(w(\boldsymbol{x})))\boldsymbol{K}^2 \tag{5}$$

where $\boldsymbol{K}$ is the cross-product matrix, only with $w(\boldsymbol{x})$ rotated.

Therefore, LMDS are defined as follows:

$$\boldsymbol{g}(\boldsymbol{x}) = \boldsymbol{M}(\boldsymbol{x})\boldsymbol{f}(\boldsymbol{x}) = \kappa\boldsymbol{R}(\boldsymbol{x})\boldsymbol{n}(\boldsymbol{x}) \tag{6}$$

where $\boldsymbol{n}(\boldsymbol{x})$ represents the state-dependent normal vector, and $\boldsymbol{R}(\boldsymbol{x})$ varies continuously in the state space.

Assuming the non-penetrability of the contact surface, the LMDS should satisfy the following relationship for all points in space:

$$\begin{cases} \boldsymbol{g}(\boldsymbol{x})^{\mathrm{T}}\boldsymbol{n}(\boldsymbol{x}) = 0 & \text{(Contact state)} \\ \boldsymbol{g}(\boldsymbol{x})^{\mathrm{T}}\boldsymbol{n}(\boldsymbol{x}) > 0 & \text{(Free space)} \end{cases} \tag{7}$$

*2.2. Learning about Locally Modulated Dynamic Systems*

The surface can be described as the set of all $\boldsymbol{x} \in \chi \subseteq \mathbb{R}^N$, where $\chi$ is the sample space. Given the surface dataset $\{(\boldsymbol{x}_1, y_1), \cdots, (\boldsymbol{x}_n, y_n)\}$ (where $n$ represents the size of the surface dataset), $\boldsymbol{x}_i \in \chi$ is the training set of surface points, and $y_i \in \mathbb{R}$ is the target value corresponding to $\boldsymbol{x}_i$. The objective of support vector regression (SVR) is to find a surface model $h(\boldsymbol{x})$ where the maximum value relative to the observed response value $y$ is $\varepsilon$, while remaining as flat as possible. The expression for the surface model $h(\boldsymbol{x})$ is as follows:

$$h(\boldsymbol{x}) = \boldsymbol{w}^{\mathrm{T}}\boldsymbol{x} + b \tag{8}$$

where $\boldsymbol{w} \in \mathcal{H}$ represents the weight vector, which is the normal vector of the hyperplane and determines the direction of the hyperplane; $b \in \mathbb{R}$ is the intercept, determining the position of the hyperplane.

The prediction of the $\varepsilon$-SVR for a given surface-point training set is as follows:

$$h(\boldsymbol{x}) = \sum_{i=1}^{m} (\alpha_i - \alpha_i^*)\boldsymbol{x}_i^{\mathrm{T}}\boldsymbol{x} + b \tag{9}$$

where $\alpha_i$ and $\alpha_i^*$ are Lagrange multipliers, with an upper bound of $C$. $\boldsymbol{x}_i \in SV \subset \boldsymbol{x}$ represents the support vectors of SVR, which are also part of the training samples, with the size of the support vector set being $m$. The resultant surface model $h(\boldsymbol{x})$ generated after training is only related to the support vectors. The constant $C > 0$ is the penalty coefficient, which controls the balance between the flatness of $f(\boldsymbol{x})$ and the permissible deviation, where the permissible deviation is less than or equal to the tolerance error $\varepsilon$. This method of controlling permissible deviation through the tolerance error is referred to as the $\varepsilon$ insensitive loss function [17].

To reduce computational costs, SVR is used to perform inner-product operations in a high-dimensional feature space by introducing a kernel function. This paper assumes the mapping of $\Phi$ from the sample space $\chi$ to a Hilbert space $\mathcal{H}$, and if there exists a

kernel function such that $K(\mathbf{x}, \mathbf{x}') = \Phi(\mathbf{x})^\text{T} \cdot \Phi(\mathbf{x}')$, then this kernel function can implicitly determine the nonlinear mapping $\Phi$. The mapping $\Phi$ defined by the kernel function transforms a 3D surface into a hyperplane in the high-dimensional Hilbert space $\mathcal{H}$.

Among all kernel functions, the Gaussian kernel function is the most commonly used. This paper's $\varepsilon$-SVR employs the following Gaussian kernel function:

$$\begin{cases} K(\mathbf{x}, \mathbf{x}') = e^{-\lambda \|\mathbf{x} - \mathbf{x}'\|^2} \\ \lambda = \frac{1}{2\sigma^2} \end{cases} \tag{10}$$

After introducing the Gaussian kernel function, the surface model can be written as follows:

$$h(\mathbf{x}) = \sum_{i=1}^{m} (\alpha_i - \alpha_i^*) K(\mathbf{x}_i, \mathbf{x}) + b \tag{11}$$

The training data of this paper include both surface training sets and additional training sets [18]. Additional training points are generated by moving given surface points $\mathbf{x}_i$ along their surface normal at a distance $y_i$. $y_i$ is distributed on both sides of the surface, with negative values below the surface and positive values above the surface. Therefore, SVR learns a hyperplane in the Hilbert space $\mathcal{H}$ through data training, which is used to estimate the distance $\Gamma(\mathbf{x})$ from any position $\mathbf{x}$ in space to the surface. The normal vector $\mathbf{n}(\mathbf{x})$ is obtained by calculating the gradient of the surface model $h(\mathbf{x})$, i.e., by taking the first-order partial derivative of $h(\mathbf{x})$:

$$\mathbf{n}(\mathbf{x}) = \frac{\partial h(\mathbf{x})}{\partial \mathbf{x}} = \sum_{i=1}^{m} (\alpha_i - \alpha_i^*) \frac{\partial K(\mathbf{x}_i, \mathbf{x})}{\partial \mathbf{x}} \tag{12}$$

## 3. Integrated Adaptive Admittance Control Strategy for Robots

### 3.1. Integral Adaptive Admittance Control

This section employs adaptive impedance control to achieve force tracking in the presence of uncertainties in the environmental information [19]. Currently, there are two main methods of adaptive impedance control: position compensation and velocity compensation adaptive impedance.

The single-degree-of-freedom control formula used for position-compensation adaptive impedance is as follows:

$$m(\ddot{x}_\text{d} - \ddot{x}_\text{c}) + b(\dot{x}_\text{d} - \dot{x}_\text{c}) + k(x_\text{d} - x_\text{c} + \Phi) = f_\text{e} - f_\text{d} \tag{13}$$

The compensation term is defined as follows:

$$\Phi(t) = \Phi(t - \lambda) + \sigma \frac{f_d(t - \lambda) - f_e(t - \lambda)}{k} \tag{14}$$

where $\lambda$ represents the sampling rate; $\sigma$ denotes the update rate; and $\sigma > 0$.

The single-degree-of-freedom control formula used for velocity compensation adaptive impedance is as follows:

$$m(\ddot{x}_\text{d} - \ddot{x}_\text{c}) + b(\dot{x}_\text{d} - \dot{x}_\text{c} + \Phi) = f_\text{e} - f_\text{d} \tag{15}$$

The compensation term is defined as follows:

$$\Phi(t) = \Phi(t - \lambda) + \sigma \frac{f_d(t - \lambda) - f_e(t - \lambda)}{b} \tag{16}$$

Equations (14) and (16) contain discrete compensation terms, making theoretical analysis difficult and requiring iterative solutions, which may lead to significant errors. Taking

position-compensation adaptive impedance control as an example, assuming an infinitesimally small sampling period, position compensation is derived based on differential theory:

$$\Phi(t + \Delta t) = \Phi(t) + \sigma\frac{f_d(t) - f_e(t)}{k} = \frac{\sigma}{k}\sum_{i=0}^{t}(f_d(i) - f_e(i))\Delta t = \frac{\sigma}{k}\int_{0}^{t}(f_d - f_e)\mathrm{d}t \quad (17)$$

Substituting Equation (17) into Equation (13) yields the control law for a single degree of freedom:

$$m(\ddot{x}_d - \ddot{x}_c) + b(\dot{x}_d - \dot{x}_c) + k(x_d - x_c) = f_e - f_d + \sigma\int_{0}^{t}(f_e - f_d)\mathrm{d}t \quad (18)$$

Based on the above derivation, extending the single degree of freedom to six degrees of freedom, we obtain the new control law:

$$M(\ddot{X}_d - \ddot{X}_c) + B(\dot{X}_d - \dot{X}_c) + K(X_d - X_c) = F_e - F_d + K_i\int_{0}^{t}(F_e - F_d)\mathrm{d}t \quad (19)$$

where $K_i$ is the integral coefficient of force error, and $X_c$ is the modified robot motion control target.

Since the new control law includes an integral term, Equation (19) is referred to as the integral adaptive impedance control law. The soft controller, designed based on the integral adaptive impedance control law, is shown in Figure 1.

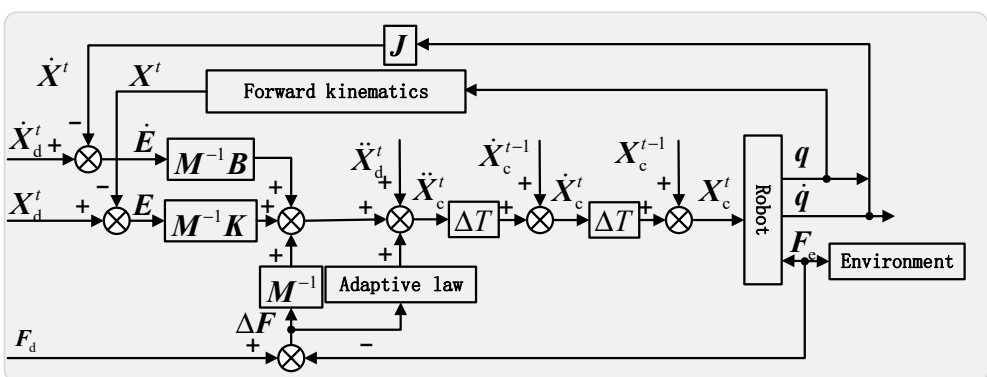

**Figure 1.** Integral adaptive admittance control block diagram.

In the soft controller, the integral term in the integral adaptive impedance control law is mainly used to compensate for system errors and external disturbances, improving the robustness and tracking performance of the control system. As the integral parameter $K_i$ increases, the overshoot will significantly increase. For convenient application in robot control, Equation (19) can be converted into a discrete format:

$$\ddot{X}_c(t) = \ddot{X}_d(t) + M^{-1}(F_d(t) - F_e(t) + K_i\sum_{\tau=0}^{t}(F_d(\tau) - F_e(\tau))T + B(\dot{X}_d(t) - \dot{X}_c(t-1)) + K(X_d(t) - X_c(t-1))) \quad (20)$$

$$\begin{cases} \dot{X}_c(t) = \dot{X}_c(t-1) + \ddot{X}_c(t)T \\ X_c(t) = X_c(t-1) + \dot{X}_c(t)T \end{cases} \quad (21)$$

where $T$ is the cycle period of the integral adaptive impedance control.

### 3.2. Stability and Convergence Analysis

Integral adaptive impedance control (IAIC) is essentially a proportional–integral control of force error, where the integral term is utilized to eliminate steady-state errors. In contrast to adaptive impedance control algorithms, it does not include discrete compensation terms, thereby allowing for the use of traditional frequency domain analysis methods

to analyze its steady-state and dynamic performance. For the sake of facilitating frequency domain analysis, the following assumptions are made:

Assumption 1: The robot can accurately track the given position, i.e., $X_c = X$.

Assumption 2: The disturbance term is the difference between the desired position and the static position of the environment, i.e., $N = X_d - X_e$.

Assumption 3: The environment follows a linear spring model, where the actual contact force depends on the spring stiffness and the displacement of the environment, i.e., $F_e = K_e(X - X_e)$.

Based on these assumptions, the outer loop of the integral adaptive impedance control can be regarded as an independent control system, where the desired force $F_d$ serves as the input, the actual contact force $F_e$ acts as the output, and the difference between the desired position and the environment position $N$ serves as the disturbance. Since the six degrees of freedom at the end of the robot are independent of each other, for simplicity, this paper focuses on the one-dimensional interaction force in the subsequent analysis. After system simplification, the block diagram of the single-degree-of-freedom integral adaptive impedance control outer-loop control system is illustrated in Figure 2.

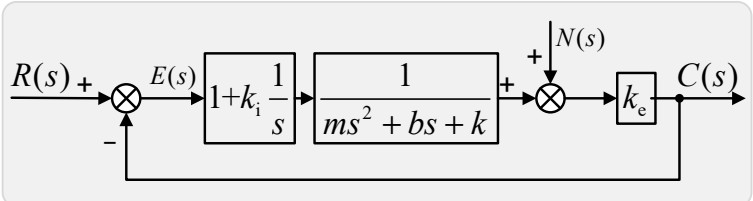

**Figure 2.** Structure diagram of integral adaptive admittance outer-loop control system.

In the figure, $R(s)$ and $N(s)$ represent external inputs to the system, $R(s)$ is the desired output contact force of the system $f_d$, $N(s)$ is the disturbance, $C(s)$ is the output contact force of the system $f_e$, and $E(s)$ is the error signal, which is the difference between the desired and actual values of the system's output contact force.

When only considering the relationship between the desired input force and the output force, by applying the superposition principle and setting $N(s) = 0$, the transfer function from the input signal $R(s)$ to the output signal $C(s)$ can be obtained:

$$\Phi(s) = \frac{C(s)}{R(s)} = \frac{k_e(s + k_i)}{ms^3 + bs^2 + (k + k_e)s + k_i k_e} \tag{22}$$

When considering only the relationship between disturbance and output force, applying the superposition principle and setting $R(s) = 0$, the transfer function from the disturbance input $N(s)$ to the output signal $C(s)$ can be obtained:

$$\Phi_n(s) = \frac{C(s)}{N(s)} = \frac{k_e(ms^3 + bs^2 + ks)}{ms^3 + bs^2 + (k + k_e)s + k_i k_e} \tag{23}$$

When both the input signal $R(s)$ and the disturbance $N(s)$ act simultaneously, the system's output is given by the following equation:

$$C(s) = \Phi(s)R(s) + \Phi_n(s)N(s) = \frac{k_e(s + k_i)}{ms^3 + bs^2 + (k + k_e)s + k_i k_e}R(s) + \frac{k_e(ms^3 + bs^2 + ks)}{ms^3 + bs^2 + (k + k_e)s + k_i k_e}N(s) \tag{24}$$

It can be seen from Equation (24) that the simplified system is a third-order linear system, and its stability can be analyzed using the Routh criterion. Since the admittance parameters of the integral adaptive impedance control system are all diagonal matrices,

only one degree of freedom of the system needs to be considered. The characteristic equation of the third-order linear system's closed-loop function is as follows:

$$ms^3 + bs^2 + (k + k_e)s + k_i k_e = 0 \tag{25}$$

The Routh table constructed according to the Routh criterion is as follows:

$$
\begin{array}{lll}
s^3 & m & k + k_e \\
s^2 & b & k_i k_e \\
s^1 & \frac{b(k+k_e) - mk_i k_e}{b} & 0 \\
s^0 & k_i k_e & 0
\end{array}
\tag{26}
$$

The first column of the Routh table should have exclusively positive elements to ensure the asymptotic stability of the third-order linear system. The admittance parameters $m$, $b$, and $k$, and environmental stiffness parameter, $k_e$ are nonnegative, and all coefficients that satisfy the characteristic equation are positive. To ensure that all elements in the first column of the Routh table are positive, it is necessary to satisfy the following equation:

$$\frac{b(k + k_e) - mk_i k_e}{b} > 0 \tag{27}$$

According to the above constraints, the constraint equation for the stability of the third-order linear system can be derived, i.e., the range of $k_i$ values is as follows:

$$0 < k_i < \frac{b(k + k_e)}{mk_e} \tag{28}$$

For a rigid environment where the environment stiffness $k_e$ is much larger than the admittance stiffness $k$, the following can be obtained:

$$0 < k_i < \frac{b}{m} \tag{29}$$

Equations (28) and (29) show that, by choosing an appropriate $k_i$ value, the asymptotic stability of the integral adaptive impedance control system can be maintained.

When input and disturbance are applied, the error signal $E(s)$ is used as the output of the integral adaptive impedance control system. Using the superposition principle, the error quantity of the system is obtained as follows:

$$E(s) = \frac{ms^3 + bs^2 + ks}{ms^3 + bs^2 + (k + k_e)s + k_i k_e} R(s) - \frac{k_e(ms^3 + bs^2 + ks)}{ms^3 + bs^2 + (k + k_e)s + k_i k_e} N(s) \tag{30}$$

For an integral adaptive impedance control system that is asymptotically stable, the input step signal is $R(s) = \frac{1}{s}$, and the steady-state error under step disturbance $e_{ss}$ is as follows:

$$e_{ss}(\infty) = \lim_{s \to 0} sE(s) = \frac{(1 - k_e)(ms^3 + bs^2 + ks)}{ms^3 + bs^2 + (k + k_e)s + k_i k_e} = 0 \tag{31}$$

When the disturbance $N(s) = \frac{1}{s}$, the steady-state error of the integral adaptive impedance control system for step input signal is zero.

To further validate the robustness of the integral adaptive impedance control algorithm, consider the case when $f_d$ is a dynamic force. Assuming the input signal is a sine function $\sin(\omega t)$, whose Laplace transform is $R(s) = \frac{\omega}{s^2 + \omega^2}$, then the steady-state error of the integral adaptive impedance control system with a step disturbance of $N(s) = \frac{1}{s}$ is as follows:

$$e_{ss}(\infty) = \lim_{s \to 0} sE(s) = \frac{(-k_e s^2 + \omega s + k_e \omega^2)(ms^3 + bs^2 + ks)}{(s^2 + \omega^2)(ms^3 + bs^2 + (k + k_e)s + k_i k_e)} = 0 \tag{32}$$

When the disturbance $N(s) = \frac{1}{s}$, the steady-state error of the integral adaptive impedance control system for sine input signal is zero.

This section analyzes the adaptability of the integral adaptive impedance control law to changes in input and disturbance and demonstrates that the algorithm has good adaptive performance. The algorithm can achieve precise robot force control and maintain stable control performance even in the face of disturbance changes.

## 4. Real-Time Motion-Planning Experiment

### 4.1. Real-Time Motion-Planning Simulation

In the previous section, a motion planner was designed based on LMDS, and the SVR algorithm with a Gaussian kernel was employed to learn the vector and distance functions of LMDS. To validate the feasibility of using LMDS to plan local polishing paths for robots, MATLAB platform was employed for path-planning simulations, utilizing the Robotics System Toolbox to define the UR3 robot model in the simulation environment.

Before conducting the path-planning simulation, the surface model was learned using the SVR algorithm with a Gaussian kernel. During the simulation process, the motion planner continuously updated the LM dynamic system based on the actual end-effector position of the robot, which required the real-time computation of LMDS. The computation process of LMDS is illustrated in Algorithm 1, with the velocity scaling factor $\kappa$ set to 0.2 and the rotation speed factor $v$ set to 20.

---

**Algorithm 1** Calculate LMDS

**Input:** $x, SV$

| | |
|---|---|
| 1: Distance | $\Gamma = \sum\limits_{i=1}^{m} (\alpha_i - \alpha_i^*)\exp(-\lambda x_i - x^2) + b$ |
| 2: Normal vector | $n = \sum\limits_{i=1}^{m} (\alpha_i - \alpha_i^*)(-2\lambda \exp(-\lambda x_i - x^2))$ |
| 3: Target vector | $v_{\mathrm{d}} = x - x^*$. |
| 4: Projection vector | $p = (I - nn^{\mathrm{T}})v_{\mathrm{d}}$. |
| 5: Potation vector | $\mu = n \times p$. |
| 6: Rotation vector | $w = (1 - \tanh(v\Gamma))\arccos\frac{n^{\mathrm{T}}p}{\|n\|\|p\|}$. |
| 7: Rotation angle | $R = I + K\sin w + K^2(1 - \cos w)$. |
| 8: Rotation matrix | $\dot{x} = \kappa R n$. |

**Output:** $\dot{x}$

---

To compute LMDS, the SVR algorithm with a Gaussian kernel ($C = 100$, $\varepsilon = 0.01$, $\sigma = 0.2$) is utilized to learn the surface model of the workpiece to be polished. The parameters generated after learning include the intercept $b$, the set of Lagrange multipliers $\alpha_i$ and $\alpha_i^*$, and the set of support vectors $SV$. Using the surface model, it is possible to obtain the normal vector $n(x)$ at any position in space and the distance $\Gamma(x)$ from any position in space to the surface.

Subsequently, the direction vector $v_{\mathrm{d}}$ is computed from any position in space to the target point $x^*$, along with the projection vector $p$ of the direction vector on the surface. Then, the rotation vector $\mu$ is calculated using the normal vector $n$ and the projection vector $p$. This is followed by the determination of the rotation angle $w$ that attenuates the rotation matrix. Finally, the rotation matrix of LMDS is computed using the Rodrigues formula, yielding the velocity vector $\dot{x}$ at any position in Cartesian space.

In the simulation, the contact surface is non-penetrable. The results of LMDS path planning simulation are illustrated in Figure 3. The motion planner updates LMDS based on the actual end-effector position of the UR3 robot arm, outputting the next desired pose and the motion velocity of UR3 relative to the current state. LMDS drives UR3 to make contact with the surface and move along it from the initial position $x_0$ to the target point $x^*$. The purple curve in the figure represents the motion trajectory of UR3.

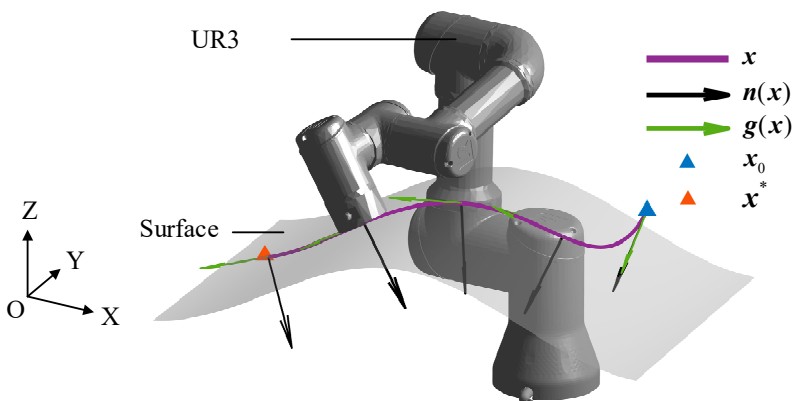

**Figure 3.** Motion-planning simulation.

Figure 4 depicts LMDS during the motion-planning simulation, including the contour levels of the surface, i.e., the distance $\Gamma(x)$ from any position in space to the surface, and the vector field of $g(x)$.

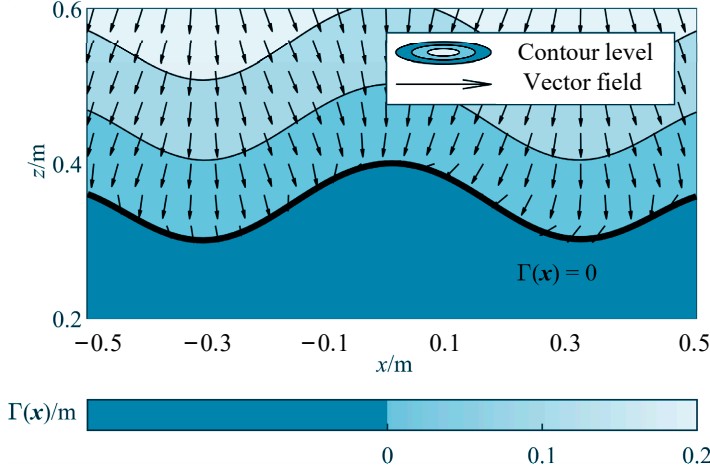

**Figure 4.** LMDS for motion-planning simulation.

In the figure, the contact surface is denoted by $\Gamma(x) = 0$, and $\Gamma(x) > 0$ represents the space above the contact surface. During the process of robot path planning with LMDS, G represents the Cartesian velocity vector, and the gray curve represents the vector field of the Cartesian velocity (i.e., the collection of Cartesian velocities $\dot{x}$). $n(x)$ represents the normal vector at the path point, which determines the desired posture of the polishing tool of the robot.

The simulation results of local robot path-planning demonstrate that LMDS ensures that the robot reaches the attractor (i.e., the target point), validating the feasibility and stability of the motion planner.

### 4.2. Local Polishing Experiment

The architecture of the polishing robot system is depicted in Figure 5, comprising a six-dimensional force sensor and a polishing tool at the robot's end effector. The control cabinet of the UR3 communicates with the PC host via a TCP/IP protocol, while the controller of the force sensor exchanges data with the PC host via USB. The PC host issues control commands to the polishing tool within the same local network via Wi-Fi, and it acquires depth image data of the workpiece from the Kinect V2 sensor via USB.

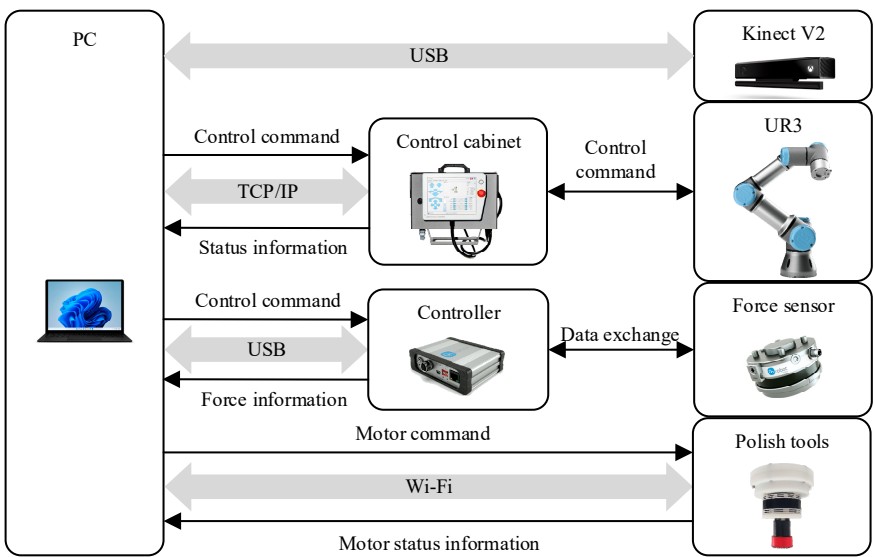

**Figure 5.** The polishing robot's system architecture.

The entire experimental setup is illustrated in Figure 6. The workpiece to be polished is a car rearview mirror, which is fixed onto the platform with a clamping plate.

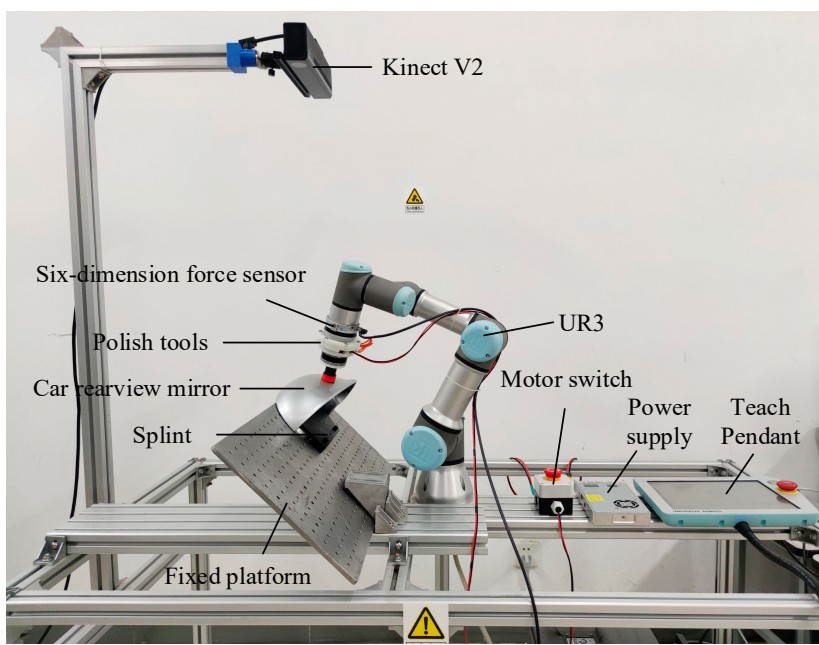

**Figure 6.** Overall experimental platform.

The purpose of this experiment is to verify the effectiveness of the proposed motion-planning algorithm. Before starting the local polishing experiment, the polishing tool is brought into contact with the surface of the car rearview mirror using the robot's teaching function. The collection of end-effector positions where the contact force is non-zero is performed to produce as a dataset for the surface to be polished. The LMDS of the car rearview mirror is learned from this dataset. Firstly, the polishing tool is brought into contact with the surface of the car rearview mirror, and the collection of end-effector positions where the contact force is non-zero produces the dataset for the surface to be polished. Then, the LMDS of the car rearview mirror is learned using the SVR algorithm with a Gaussian kernel. Finally, the LMDS drives the UR3 robot arm to make contact with the surface of the car rearview mirror and move from the initial position to the target point.

The polishing experiment process is illustrated in Figure 7. Before polishing, black marker pen traces were visible on the surface of the car rearview mirror, as shown in (a) of Figure 7. After polishing, the black marker pen traces were completely removed, as shown in (b) of Figure 7.

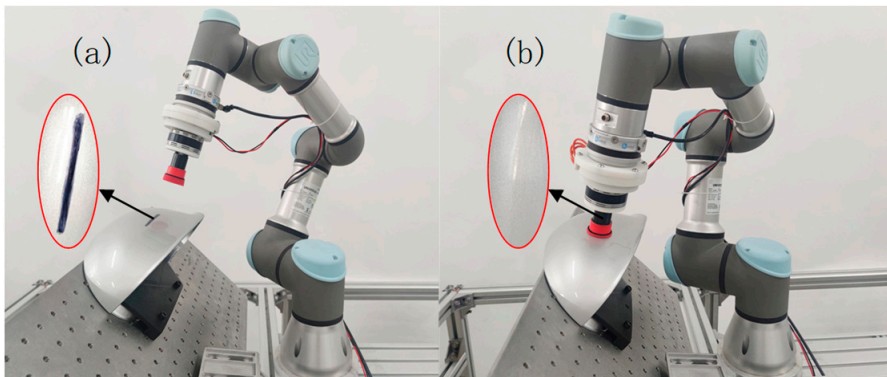

**Figure 7.** Polishing experiment process. (**a**) Surface condition before polishing; (**b**) Surface condition after polishing.

The trajectory of the polishing tool's end effector during the local polishing experiment is shown in Figure 8. The black dots represent the local surface dataset collected before polishing, and the gray curve represents the trajectory of the polishing tool's end effector generated by the LMDS. The robotic arm moves the polishing tool from the initial position $x_0(0.356, -0.100, 0.37)$ to the target point $x^*(0.356, 0.040, 0.315)$.

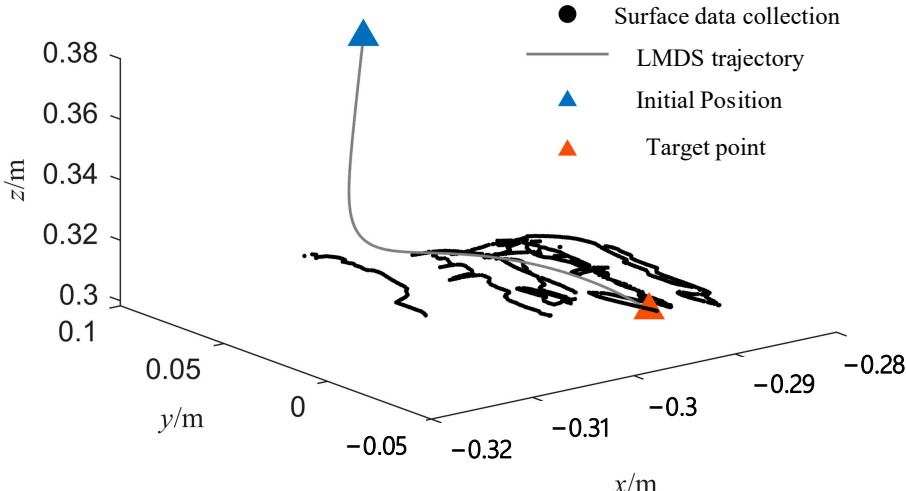

**Figure 8.** Local polishing experiment's polishing-tool end trajectory.

During the local polishing experiment, the contact force in the Z direction of the polishing tool's coordinate system is collected using the six-dimensional force sensor, with a sampling period of 8 ms. The actual change in the contact force between the polishing tool and the car rearview mirror surface is shown in Figure 9. The gray solid line represents the contact force in the Z direction of the polishing tool's coordinate system, fluctuating around $-10 \pm 1.5$ N. It can be observed from the figure that the integral adaptive impedance controller effectively suppresses the impact between the polishing tool and the car rearview mirror during contact and ensures a constant contact force between them.

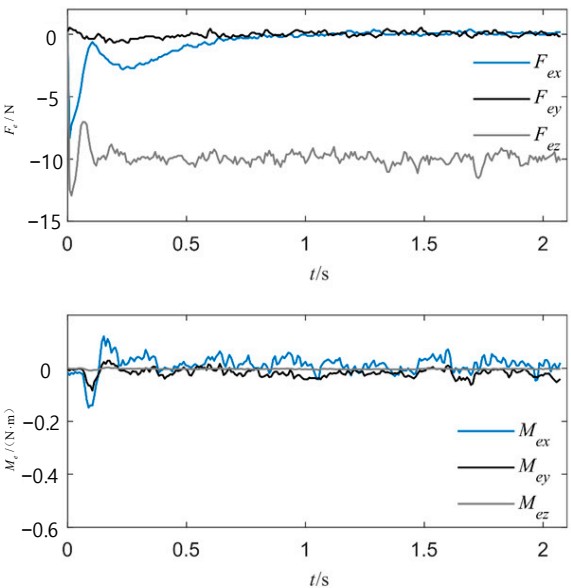

**Figure 9.** Contact forces in local polishing experiments.

The criteria for evaluating polishing quality primarily relate to the geometric morphology, physical properties, and chemical properties of the polished surface. Surface roughness is one of the most direct and commonly used indicators of polishing quality; a lower surface roughness means a smoother surface. Therefore, two indicators, namely the average surface roughness (Ra) and the maximum profile height (Rz), are used to describe surface roughness and evaluate the polishing effect. The same position on the surface before and after polishing is scanned using the Keyence laser profiler LJ-X8020 to obtain surface information. Figure 10 shows the surface texture and profile of the same local position on a car rearview mirror surface before and after polishing. The original surface was only processed with a polishing cloth, with a roughness Ra of 0.27 μm and an Rz of 2.02 μm. After polishing, the surface roughness Ra decreased to 0.042 μm, and the Rz decreased to 0.25 μm. The effective reduction in the surface roughness of the car rearview mirror once again confirms the effectiveness and feasibility of the designed motion-planning method for use in the polishing robot system, and it is suitable for polishing freeform surfaces.

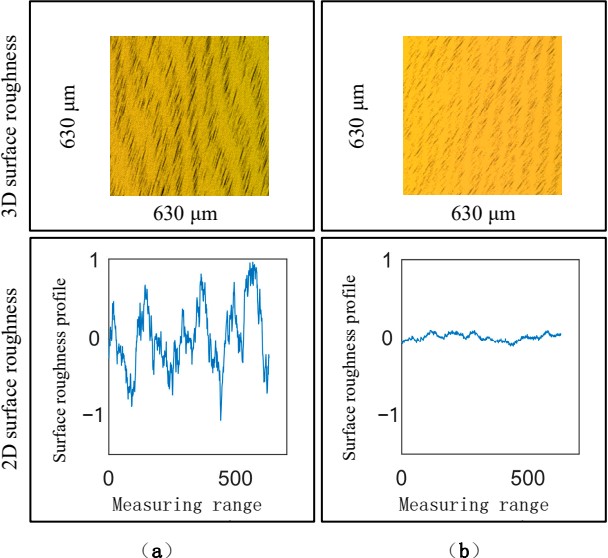

**Figure 10.** Surface roughness of the automobile rearview mirror. (**a**) Surface roughness before polishing; (**b**) Surface roughness after polishing.

## 5. Discussion

Through path-planning simulations and polishing experiments, the feasibility and stability of motion planning are validated based on locally modulated dynamic systems for real-time path generation. This approach meets the requirements for precise force control, making it applicable in practical polishing tasks.

The integral adaptive admittance control algorithm proposed in this paper is implemented based on the position control of the robot, and the accuracy of force control depends on the repeatability of the robot's positioning. In practical production, high polishing efficiency is required, necessitating further improvement in the frequency of the integral adaptive admittance controller. For instance, reducing the control cycle to 1 ms or utilizing the velocity control loop of the robot to achieve force control could be explored.

## 6. Conclusions

The paper proposes a trajectory planning method based on dynamic systems for generating trajectories during polishing processes. It utilizes a six-dimensional force sensor to measure contact forces and torques, integrating them with integral adaptive impedance control to maintain constant contact forces. This strategy, rooted in the local modulation of dynamic systems, offers flexible and smooth motion and force generation, ensuring that the robot reaches the target surface and moves smoothly along it, an attribute which can be applied to real polishing work.

However, the polishing experiments performed in this study did not delve into specific polishing techniques. In future research, various polishing discs, abrasives, and contact pressures could be employed to polish automotive rearview mirrors, thereby establishing a comprehensive set of automotive rearview mirror-polishing techniques. Moreover, during the polishing process, the robotic polishing system solely relied on force information as determined using a six-axis force sensor. Subsequent studies could explore the utilization of a multi-sensor fusion approach to devise a robotic polishing system. For instance, researchers could integrate sensors such as vision and laser sensors for the real-time surface quality assessment of automotive rearview mirrors. Subsequently, based on the information obtained throudgh sensor fusion, an expert system could be employed to adjust polishing parameters, thereby enhancing the quality of the polishing process.

**Author Contributions:** X.W. (Xinqing Wang) contributed to the conception of the study, and helped perform the analysis with constructive discussions. X.W. (Xin Wang) performed the experiment, and contributed significantly to analysis and manuscript preparation. Z.Y. performed the data analyses and wrote the manuscript. Y.Z. reviewed the manuscript and data. All authors have read and agreed to the published version of the manuscript.

**Funding:** This research was funded by the Shandong Provincial Major Science and Technology Innovation Project and the Shandong Provincial Natural Science Foundation Project, grant number 2017CXGC0902 and ZR2022MF291, respectively.

**Data Availability Statement:** No new data were created or analyzed in this study. Data sharing is not applicable to this article.

**Acknowledgments:** The authors would like to thank the anonymous reviewers for the insightful comments and valuable suggestions.

**Conflicts of Interest:** The author declares no conflicts of interest.

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
