# Peer review of "An Investigation of Real-Time Robotic Polishing Motion Planning Using a Dynamical System"

_machines, doi:10.3390/machines12040278_

Round 1

Reviewer 1 Report

Comments and Suggestions for Authors

1. Generic comments:

The title is clumsy, its does reflect the content, but needs rewording. Maybe something such as:

“An investigation of Real-Time Robotic Polishing Motion Planning Using a Dynamical System”

Abstract is appropriate and reflects article content.

‘Sports Planning’ should be removed from the keywords

Although iThenticate shows a high 21% commonality, after reviewing the report, the software has picked up very generic terms and phases, and should therefore be ignored.

The literature review is quite old, there has been a large volume of research in robotic polishing that should be acknowledged.

Could figure 5 be produced in colour to aid readability.

2. Technical comments

What happens if assumptions 1 to 3 are not met? Is there any contingency in the model if the spring model fails?

Need to define why Ra and Rz are being used to understand surface texture?

The discussion section just summarises elements, this needs to provide some form of dialogue around the success of the approach, limitations identified and potential rectifications.

Article requires a conclusion; it currently stops abruptly after the discussion. There needs to be some form of future works acknowledged, what is the next phase of the investigation and who would benefit from the work?

Reviewer 2 Report

Comments and Suggestions for Authors

Dear authors, congratulations on your interesting work and, above all, on the final results. The article concerns an important, practical problem. This is an issue currently being addressed in the industry, so I suggest slightly expanding the literature review with more recent works. Similar issues regarding polishing in contact with a flexible environment are also analyzed.

In this work also lacked a systematic description of the introduced variables, which makes a systematic analysis of mathematical expressions difficult. For example, the variable μ appears on line 104, and on line 303 we have a rotation vector defined. These equation descriptions should be reviewed and corrected.

I evaluate the article itself positively, especially since the experimental results confirmed the high effectiveness of both force stabilization and polishing quality.

Reviewer 3 Report

Comments and Suggestions for Authors

The authors design a dynamic system-based contact motion planning strategy for robot polishing and sanding to produce trajectories during localized polishing: an overall adaptive impedance control strategy is used to achieve flexible control of the robot. However, the following problems still exist:

1. The title of the paper uses the term "Dynamical System", while the text uses the term "dynamic systems", which should be carefully considered by the author?

2. In the article, the Dynamic Systems method is used to solve the motion path planning problem for polished objects, but the information such as the path and normal vector of the car's rearview mirror is not seen in the experimental Fig. 9?

3. What are the advantages of the method used in this article compared to end mounted pneumatic constant force devices?

Comments on the Quality of English Language

1. The title of the paper uses the term "Dynamical System", while the text uses the term "dynamic systems", which should be carefully considered by the author?
